

# Effects of a six-week weighted-implement throwing program on baseball pitching velocity, kinematics, arm stress, and arm range of motion

Joseph A. Marsh[1,*], Matthew I. Wagshol[1,*], Kyle J. Boddy[1], Michael E. O'Connell[1], Sam J. Briend[2], Kyle E. Lindley[1] and Alex Caravan[1]

[1] Research and Development, Driveline Baseball, Inc, Kent, WA, United States of America
[2] High Performance, Driveline Baseball, Inc, Kent, WA, United States of America
[*] These authors contributed equally to this work.

## ABSTRACT

**Background.** Weighted-baseball training programs are used at the high school, collegiate, and professional levels of baseball. The purpose of this study was to evaluate the effects of a six-week training period consisting of weighted implements, manual therapy, weightlifting, and other modalities on shoulder external rotation, elbow valgus stress, pitching velocity, and kinematics.

**Hypothesis.** A six-week training program that includes weighted implements will increase pitching velocity along with concomitant increases in arm angular velocities, joint kinetics, and shoulder external rotation.

**Methods.** Seventeen collegiate and professional baseball pitchers (age range 18–23, average: $19.9 \pm 1.3$) training at Driveline Baseball were evaluated via a combination of an eight-camera motion-capture system, range-of-motion measurements and radar- and pitch-tracking equipment, both before and after a six-week training period. Each participant received individualized training programs, with significant overlap in training methods for all athletes. Twenty-eight biomechanical parameters were computed for each bullpen trial, four arm range-of-motion measurements were taken, and pitching velocities were recorded before and after the training period. Pre- and post-training period data were compared via post-hoc paired $t$ tests.

**Results.** There was no change in pitching velocity across the seventeen subjects. Four biomechanical parameters for the holistic group were significantly changed after the training period: internal rotational velocity was higher (from $4,527 \pm 470$ to $4,759 \pm 542$ degrees/second), shoulder abduction was lower at ball release ($96 \pm 7.6$ to $93 \pm 5.4°$), the shoulder was less externally rotated at ball release ($95 \pm 15$ to $86 \pm 18°$) and shoulder adduction torque was higher (from $103 \pm 39$ to $138 \pm 53$ N-m). Among the arm range of motion measurements, four were significantly different after the training period: the shoulder internal rotation range of motion and total range of motion for both the dominant and non-dominant arm. When the group was divided into those who gained pitching velocity and those who did not, neither group showed a significant increase in shoulder external rotation, or elbow valgus stress.

**Conclusions.** Following a six-week weighted implement program, pitchers did not show a significant change in velocity, joint kinetics, or shoulder external rotation range of motion. When comparing pitchers who gained velocity versus pitchers who did not,

Corresponding authors
Joseph A. Marsh,
joe@drivelinebaseball.com,
jamarsh123@gmail.com
Matthew I. Wagshol,
mwagshol@drivelinebaseball.com

no statistically significant changes were seen in joint kinetics and shoulder range of motion.

# INTRODUCTION

Studies on underweight and overweight baseballs have shown a positive training effect on the throwing velocity of regulation-weight baseballs (*Derenne & Szymanski, 2009*; *DeRenne, Ho & Blitzblau, 1990*; *Egstrom, Logan & Wallis, 1960*; *DeRenne, 1985*). Additionally, studies have also shown no negative effects of throwing underweight and overweight implements on pitching control or injury risk (*Derenne & Szymanski, 2009*; *DeRenne et al., 1994*).

A recent biomechanical study shows that pitching slightly underweight and overweight baseballs can produce variations in kinematics (specifically arm, trunk, pelvis, and shoulder velocities) without increased arm kinetics (*Fleisig et al., 2015*) and that maximum-effort crow-hop throwing with the same implements can increase shoulder internal rotation angular velocity and elbow varus torque (*Fleisig et al., 2017a*; *Fleisig et al., 2017b*). Additionally, there have been indications that weighted-baseball throwing can increase shoulder external rotation in a six-week training period on high school athletes (*Reinold et al., 2004*).

There have also been investigations into heavier-weighted plyometric throws used in training and rehab programs, including but not limited to two handed chest passes and side throws of 8-pound "plyoballs" or the more traditionally-named medicine balls (*Wilk, Meister & Andrews, 2002*). Eight weeks of plyometric training can increase shoulder internal rotation power and throwing distance (*Fortun, Davies & Kernozck, 1998*). A different study using plyoballs and "The Ballistic Six" found a significant increase in throwing velocity (*Carter et al., 2007*). While there is also research suggesting that throwing weighted plyos from 2–8 lb. may improve proprioception (*Swanik et al., 2002*).

Driveline baseball (Seattle, Washington, USA) has developed weighted baseball training programs, which have been used by many professional and collegiate pitchers. Those pitchers who completed the weighted implement training programs have on average increased pitching velocity 2.7 MPH in 2016 and 3.3 MPH in 2017 (https://www.drivelinebaseball.com/2016/08/college-summer-wrap/ and https://www.drivelinebaseball.com/2017/09/driveline-baseball-review-college-summer-training-results-2017/). However, there remains no conclusive evidence explaining the mechanism of the velocity increase, and research indicates the phenomenon of weighted-ball training increasing "arm strength" may be incorrect (*Cressey, 2013*).

Increases in throwing shoulder external rotation and loss of throwing shoulder internal rotation are potentially deleterious (*Wilk et al., 2011*), but, to our knowledge, no weighted-implement training program combines a throwing program with other training modalities to potentially reduce negative adaptive effects on the arm. There is evidence that certain

mobility programs can reduce the negative adaptive effects of throwing that lead to arm fatigue, loss of strength and/or injury (*Laudner, Sipes & Wilson, 2008*), and it is theorized that heavy resistance training and manual therapy may aid in this regard.

The purpose of this study was to evaluate the training effects of a weighted-implement throwing program that includes individualized training routines focused around combating the negative effects of throwing on pitching velocity, external rotation and elbow varus torque. We hypothesize the previously described program will increase external rotation, ball velocity, and elbow varus torque.

## METHODS

### Participants and informed consent

Healthy and asymptomatic college and professional pitchers were recruited from the Driveline Baseball 2017 training group via opt-in forms. Prior to being included in the study, investigators asked the pitchers about their current injury status. Pitchers were excluded if they had current symptoms of arm or shoulder pain or fatigue, or any other pain or discomfort that would prohibit completion of the study. Additionally, a prerequisite to train in the Driveline Baseball spring-summer group required medical clearance and a certified athletic trainer's sign-off before throwing pitches off a mound. Pitchers were not excluded based on previous history of injuries that did not currently manifest themselves. Pitchers were not excluded based on previous training history, although a few had trained at Driveline Baseball right before the study and most had experimented remotely with Driveline methods; the average time spent at Driveline right before the study's start was $16 \pm 10$ days, with a maximum of 41 and a minimum of 3 days.

Pitchers were scheduled to come into the Driveline Baseball Research Facility (Kent, WA) pre-testing. Upon arrival, participants were provided a verbal explanation of the study and asked to read and sign an Informed Consent document before beginning. The investigator verbally confirmed the Informed Consent document in addition to obtaining a witnessed, legal signature from the pitcher, only proceeding if the pitcher submitted both a valid signature *and* verbally confirmed acceptance of all the risks contained within the Informed Consent document.

The study was approved by Hummingbird IRB, who granted ethical approval to carry out the study at the author's facilities (Hummingbird IRB #: 2017-29, Protocol WB-DLR-115).

Twenty-one baseball pitchers (age range: 18–23) with at least high school and college pitching experience met these criteria and agreed to participate. Four were excluded bringing the final number to seventeen. The data on these pitchers is recorded in Table 1.

### Range of motion testing

During the testing period, range of motion measurements were taken using a goniometer to measure shoulder internal and external rotation in both the dominant and non-dominant arms. The same investigator was used for each individual in the initial and final tests; previous research has shown high intra-reliability for goniometer measurements (*Boone et al., 1978*). Each pitcher was measured on the same day as their motion capture-based biomechanical screening discussed below.

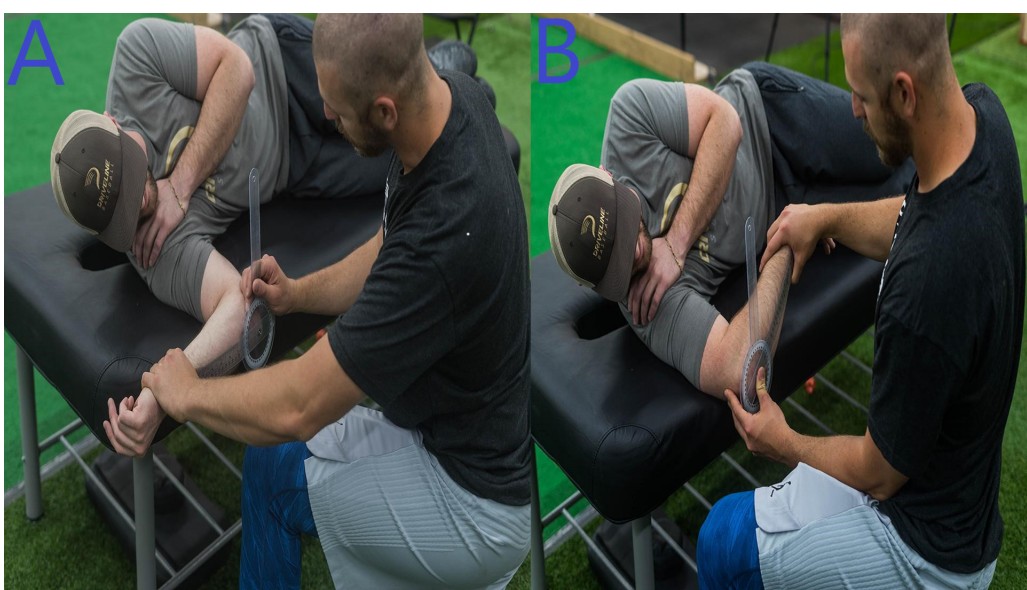

**Figure 1** **Range of motion testing.** Photo credit: Marques Gagner.

**Table 1** **Athlete data.** General measurement data on the athletes in this study like weight/height etc.

| Age (Yrs) | Height (CM) | Weight Pre (KG) | Weight Post (KG) |
|---|---|---|---|
| Age: $19.9 \pm 1.3$ | $184.8 \pm 5.0$ | $88.6 \pm 6.3$ | $88.98 \pm 6.2$ |

Measurements were taken with each athlete lying in the lateral decubitus position (Fig. 1). Testing was done in this position due to the fact that when lying supine, the humeral head is more likely to glide forward in the glenoid causing irritation in the anterior shoulder and leading to more inaccurate measurements as the athlete can compensate for a lack of range of motion through anterior or posterior rotation of the shoulder. In the lateral decubitus plane, the humeral head is in a more advantageous position to externally (Fig. 1A) and internally rotate (Fig. 1B) without humeral head glide (*Reinold et al., 2004*).

The investigator performing this part of the study was a certified strength and conditioning coach with seven years of experience and specifically trained in measuring range of motion of the shoulder using standard tools. Once the athlete was in the appropriate position, the investigator passively moved the arm until tension was reached and the measurement was taken. The intraclass correlation coefficient (ICC) of a trained clinician performing total range of motion tests of the shoulder have shown to be very reliable (*Wilk et al., 2009*).

## Kinematics

The pitchers threw as many warm-up pitches as they liked prior to beginning. Next, pitchers were fitted with reflective markers in preparation for three-dimensional motion capture. Forty-eight reflective markers were attached bilaterally on the third distal phalanx, lateral and medial malleolus, calcaneus, tibia, lateral and medial femoral epicondyle, femur,

anterior and posterior iliac spine, iliac crest, inferior angle of scapula, acromial joint, midpoint of the humerus, lateral and medial humeral epicondyle, midpoint of the ulna, radial styloid, ulnar styloid, distal end of index metacarpal, parietal bone, and frontal bone, as well as on the C7 and T10 vertebrae, the sternal end of the clavicle, and the xiphoid process.

Pitchers then threw between 3–8 maximum effort throws, with approximately 30–60 s of rest between pitches, in order to ensure enough appropriate takes captured on the motion capture system for appropriate analysis. Fatigue was assumed to be negligible with such a low pitch count. Throws were made using a 5-oz. (142g) regulation baseball off the mound to a strike zone target (Oates Specialties, LLC, Huntsville, TX, USA) located above home plate, which was $60'6''$ (18.4 m) away.

## Testing preparation

For each trial, ball velocity was measured by a Doppler radar gun (Stalker Radar; Applied Concepts, Richardson, TX, USA). Three-dimensional kinematics were tracked using an 8-camera automated motion-capture system, sampling at 240 Hz (Prime 13 System, Natural Motion/Optitrack, Corvallis, Oregon), shown in research to be comparable to more commonly-used high-end motion-capture systems (*Thewlis et al., 2013*). Cameras were placed symmetrically around the capture volume, approximately 2.4 m from the center of the pitching mound, at roughly 2.4 m high.

Testing concluded when the investigators were satisfied they had recorded three successful throws for analysis. Sample photographs and high-speed videos (Sanstreak Corp., San Jose, CA, USA) of the setup and pitches are shown in Supplementary Information.

## Biomechanical data analysis

In total, 28 kinematic and kinetic measures (11 position, 6 velocity, and 11 kinetic) were calculated using the ISB recommended model of joint coordinate systems (*Wu et al., 2002*) with code based on Fleisig methods (*Fleisig et al., 2017a*; *Fleisig et al., 2017b*) in Visual3D (C-Motion Inc., Germantown, MD, USA). Marker position data was filtered using a 20-Hz fourth-order Butterworth low-pass filter. The mean values for all variables were calculated for each participant from their 3 clearest throws, which were based upon marker and motion readability (*Escamilla et al., 1998*).

Five joint angles were calculated at the events of foot contact (FC) and ball release (BR), including: elbow flexion, shoulder horizontal abduction, shoulder abduction, shoulder external rotation, and wrist extension. Additionally, maximum dynamic shoulder external rotation was measured. All kinematic measures were all taken as their local joint angles, using local coordinate systems.

Six velocity parameters included pelvis angular velocity at FC and BR, and maximum pelvis angular velocity, upper torso angular velocity, elbow extension angular velocity, shoulder internal rotation angular velocity. Pelvis and upper torso angular velocities were measured as rotations in the global coordinate system. Elbow, shoulder, and wrist velocities were calculated as the rate of change in the joint angle and is expressed as °/s.

Maximum elbow and shoulder kinetics were calculated as either a force or a torque applied to the joint by the proximal segment onto the distal segment. Six forces were

calculated: medial, anterior, and compression distraction forces on the elbow, and superior, anterior, and compression distraction forces on the shoulder. Five joint torques were computed through inverse dynamics of the kinematic values: elbow flexion torque, elbow varus torque, shoulder horizontal adduction torque, shoulder adduction torque, and shoulder internal rotation torque, based off the resultant shoulder internal rotation moment.

## Training methods

In between pre and post tests, pitchers were exposed to a six-week training program, slightly individualized for each athlete based on their strengths and weaknesses, which was determined from their biomechanical and performance assessment. Pitchers were placed into one of three different categories for throwing programming. These were velocity development, mound development, or a hybrid version of the two. All athletes performed their training program six days a week with the seventh day being an off day.

### Warm-up

Each pitcher began a warm-up using foam rollers and lacrosse balls for self-myofascial release (SMR) of various lower body and throwing arm muscles. Another option was rolling out the forearm with Arm Aid Extreme devices (The Armaid Company, Inc., Blue Hill, ME). Athletes were allowed to SMR for a period of time that they determined necessary and were able to use SMR on other body parts if necessary. The standard SMR exercises can be found in the Supplemental Information 6.

Following SMR, athletes completed a set of exercises using Jaeger Band surgical tubing (Jaeger Sports, Los Angeles, CA, USA). Pitchers performed a forward fly to overhead reach, reverse fly to overhead reach, bicep curl with supination, tricep extension with pronation, internal and external rotations with elbow at shoulder height. Further details on the exercises can be found on pages 8–12 of the supplemental materials of HTKC1

Although Jaeger bands use a wrist cuff, surgical-tubing exercises with a handle have been shown to result in low to moderate EMG activation of the rotator cuff and surrounding musculature (Myers et al., 2005). Surgical tubing exercises can improve velocity and shoulder internal and external strength (Baheti, 2000).

Following band work, pitchers performed a series of exercises with an Oates Specialties shoulder tube (Oates Specialties LLC, Hunstsville, TX). The tube is intended for oscillation work to warm up the rotator cuff muscles. Pitchers performed shoulder flexion, shoulder abduction, external/internal rotations, pronation/supination, and stride-length forward shoulder rotations. More detail on these exercises can be found on pages 13–16 of Supplemental Information 6.

The pitchers then performed a series of four exercises with 4.5 kg wrist weights. The goals of these exercises were to warm up the muscles of the forearm and the posterior of the shoulder eccentrically. The exercises were Pronated Swings (with two-arms), Two-Arm Throws, modified Cuban Press, and Pivot-Pickoff Throws. Further details of the exercises can be found on pages 17–24 in Supplemental Information 6.

### Weighted-ball training

Athletes then moved to a specific series of throws using plyometric PlyoBalls (custom made soft sand-filled weighted balls ranging from 100–2,000 grams). There were five exercises performed; each exercise was unique within the constraints of the body's position to focus on different mechanical elements. Pitchers performed Reverse Throws, Pivot Pickoffs, Roll-in Throws, Rockers, and Walking Windups.

The ball weights, sets, and reps were all standard across the participants, depending on the training day. Pitchers completed the above warm-up six days a week with the volume and intensity of PlyoCare throws varying on the day. The throwing schedules and explanations on how to perform the exercises are listed on pages 25–36 in Supplemental Information 6.

### Long toss

On hybrid days, touched upon below, pitchers were scheduled to long-toss. Two different types of long toss days were implemented. The first was a lower intensity day. Rate of perceived exertion (RPE) was around 60–70% for the athlete accompanied by loose, relaxed throwing with a large arc as the athlete backs up in distance. Maximum distance was determined by throwing ability and RPE and as such will vary from athlete to athlete. This day did not include any high intensity compression throws.

The second type of long toss day was similar to the first except performed at an RPE of 80–90% and the athlete carries the extension throws out to maximum throwing distance. Upon reaching maximum throwing distance in as many or as few throws as required, the athlete performs eight to twelve high intensity compression throws. These compression throws remove the arc from the throw and are thrown roughly parallel to the ground from the throwers release point. Number of throws will vary day to day for each individual athlete as they are instructed to be receptive to their body's response and personal comfort level.

Research on long-toss has largely focused on throws at max distance while throwing hard on-a-line, with one study finding max distance throws resulted in more torque than in pitching (*Fleisig et al., 2011*). Another study found that max distance, hard on-a-line throws resulted in similar loads to pitching (*Slenker et al., 2014*).

Long-toss as described in the programming did not solely consist of max distance, hard on-a-line throws. Most consisted of high-arc (extension) throws to a tolerable distance for the day, otherwise described as catch-play to a distance that is tolerable. Certain training days did consist of hard on-a-line (compression) throws, which are marked in the Supplemental Information. It is important to note these distinctions since a recent study showed that many coaches, ATCs, and players define long-toss differently (*Stone et al., 2017*).

### Post-workout recovery

Each pitcher completed a post-throwing exercise circuit after each day of throwing workouts. The circuit consisted of standing rebounders; the pitchers threw a 4- and 2-lb. PlyoCare ball at a trampoline on the ground and were told to "stick" the catch of the ball or stop its upward momentum right away.

Next, were reverse scapular pull-aparts, anterior band pull-aparts, and the no money drill. After band exercises, pitchers performed waiter walks. The pitchers held a kettlebell so that their humerus lined up at shoulder height with the shoulder flexed to ninety degrees and the forearm facing vertically while walking. The kettlebell was gripped by the handle with the weight facing the ceiling. More details of the post throwing circuit can be found in Supplemental Information 6.

After the exercise circuit, each pitcher used the Marc Pro EMS device (Marc Pro, Huntington Beach, CA). The Marc Pro has been shown to improve muscle performance, recovery, and reduce Delayed Onset Muscle Soreness (DOMS) caused by exercise (*Westcott et al., 2011*; *Westcott et al., 2013*). It has been proposed that these results come from an increase in blood flow (*DiNubile et al., 2011*).

### Strength and conditioning training

In conjunction with the throwing program athletes were involved in a strength and conditioning program. This program included lifting weights, medicine ball throws, and mobility work. This program was individualized to each athlete depending on a separate physical and athletic screening.

Pitchers saw a physical therapist during the training period. Trainers are also certified in Functional and Kinetic Treatment with Rehabilitation (FAKTR), cupping, and other manual therapy techniques. Athletes were able to receive treatment on an as-needed basis.

Each pitcher had five- to six-throwing days scheduled a week. The throwing days were classified as high-intent days, hybrid days (medium intent days), and recovery days (low intent days), with the intensity and volume of throws changing per day. Athletes typically performed two high intensity days, one moderate intensity day and three recovery days within a given seven day cycle.

### Statistical analysis

To be included in the post data collection, pitchers had to participate in at least 90% of the training days. Four of the twenty-one pitchers initially chosen for the failed to meet this criterion.

Data from the training periods—including schedules, workloads, lifting programs, and intermediate progress—can be found in the Supplemental Information as spreadsheets for all pitchers.

All statistical analyses were performed using R (*RStudio Team, 2018*). After collecting and preprocessing each individual athlete's data (the initial and post biomechanical parameters, the range of motion measurements, and velocity data), means and standard deviations were calculated for each measure, and then the differences were computed, along with the subsequent t metric and $p$-value. A paired $t$-test was used due to a relatively small sample size and unknown true population variance, as data was not collected from the larger population of pitchers at Driveline. To calculate the $t$ metric, the mean differences between observations were divided by the standard error of these differences, which was calculated by the standard deviation of differences divided by the square root of the sample size, $n$. An $n - 1$ degree of freedom was used, along with an alpha level of 0.05, leaving the pure probabilistic chance of any metric being highlighted as a false positive as 5% or

**Table 2  ROM data pre/post.** Shoulder range of motion data pre/post training period.

| | Pre-test ($n = 17$) | Post-test ($n = 17$) | P Value for Pre and post-test comparison |
|---|---|---|---|
| Dominant arm internal ROM (°) | 53 ± 13 | 60 ± 15 | 0.006[*] |
| Dominant arm external ROM (°) | 122 ± 21 | 123 ± 10 | 0.637 |
| Dominant arm total ROM (°) | 174 ± 21 | 184 ± 16 | 0.031[*] |
| Non-dominant arm internal ROM (°) | 66 ± 13 | 79 ± 11 | <0.001[*] |
| Non-dominant arm external ROM (°) | 107 ± 17 | 107 ± 14 | 0.990 |
| Non-dominant arm total ROM (°) | 173 ± 17 | 185 ± 15 | 0.013[*] |

**Notes.**
*indicates that value was found to be statistically significant.

**Table 3  Shoulder ROM, velocity increase group.** Shoulder range of motion data for those who gained velocity.

| | Pre-test ($n = 9$) | Post-test ($n = 9$) | P Value for pre and post-test comparison |
|---|---|---|---|
| Dominant arm internal ROM (°) | 55 ± 14 | 64 ± 17 | 0.056 |
| Dominant arm external ROM (°) | 119 ± 25 | 122 ± 11 | 0.648 |
| Dominant arm total ROM (°) | 174 ± 23 | 186 ± 19 | 0.124 |
| Non-dominant arm internal ROM (°) | 64 ± 11 | 77 ± 10 | 0.005[*] |
| Non-dominant arm external ROM (°) | 106 ± 8 | 113 ± 12 | 0.031[*] |
| Non-dominant arm total ROM (°) | 169 ± 11 | 191 ± 15 | 0.002[*] |

**Notes.**
*indicates that value was found to be statistically significant.

less. A post-hoc analysis with similar statistical methods was also performed on both the subgroup of pitchers who saw a velocity increase during the training period and those who saw a velocity decrease.

## RESULTS

Pre- and post-range of motion tests are shown in Table 2. Four arm range of motion measurements were significantly different after the training period: internal rotation range of motion and total range of motion were *both* significantly higher for *both* dominant and non-dominant arms. Shoulder external-rotation range of motion did *not* change significantly after the training period.

Splitting the groups into pitchers that gained velocity and those who did not gain velocity did not yield significant differences between the groups. For instance, when those who gained throwing velocity were split into their own group ($n = 9$) the gain in post-training passive shoulder external-rotation range of motion was 2.8 ± 9.0 degrees, which was not statistically significant.

Range-of-motion changes of the increase and decrease velocity groups can be found in Tables 3 and 4 below.

**Table 4   Shoulder ROM, velocity decrease group.** Shoulder range of motion data for those who lost velocity in the training period.

| | Pre-test ($n = 8$) | Post-test ($n = 8$) | P Value for pre and post-test comparison |
|---|---|---|---|
| Dominant arm internal ROM (°) | 50 ± 13 | 57 ± 13 | 0.062 |
| Dominant arm external ROM (°) | 125 ± 17 | 125 ± 10 | 0.895 |
| Dominant arm total ROM (°) | 175 ± 19 | 182 ± 13 | 0.133 |
| Non-dominant arm internal ROM (°) | 68 ± 16 | 80 ± 12 | 0.049[*] |
| Non-dominant arm external ROM (°) | 108 ± 25 | 99 ± 14 | 0.360 |
| Non-dominant arm total ROM (°) | 176 ± 23 | 180 ± 13 | 0.636 |

**Notes.**
*indicates that value was found to be statistically significant.

**Table 5   Kinematic data on all athletes.** Kinematic/biomechanical data on all athletes in the study.

| | Pre-test ($n = 17$) | Post-test ($n = 17$) | P Value for pre and post-test comparison |
|---|---|---|---|
| *Front foot contact* | | | |
| Elbow flexion (°) | 91 ± 28 | 93 ± 20 | 0.651 |
| Shoulder horizontal abduction (°) | 44 ± 14 | 46 ± 18 | 0.407 |
| Shoulder abduction (°) | 83 ± 12 | 86 ± 13 | 0.163 |
| External rotation (°) | 33 ± 23 | 30 ± 23 | 0.444 |
| Wrist extension (°) | 20 ± 19 | 18 ± 19 | 0.221 |
| Pelvis angular velocity (°/s) | 92 ± 10 | 93 ± 9 | 0.475 |
| *Arm cocking/ acceleration phase* | | | |
| Maximum pelvis angular velocity (°/s) | 733 ± 104 | 721 ± 145 | 0.586 |
| Maximum torso angular velocity (°/s) | 966 ± 96 | 998 ± 103 | 0.129 |
| Maximum internal rotation velocity (°/s) | 4,527 ± 470 | 4,759 ± 542 | 0.013[*] |
| Maximum elbow extension velocity (°/s) | 2,230 ± 227 | 2,270 ± 328 | 0.499 |
| Maximum External Rotation (°) | 168 ± 10 | 167 ± 9 | 0.654 |
| *Ball release* | | | |
| Elbow flexion (°) | 16 ± 6 | 17 ± 6 | 0.526 |
| Shoulder horizontal abduction (°) | 0 ± 7 | 1 ± 7 | 0.320 |
| Shoulder abduction (°) | 96 ± 8 | 93 ± 5 | 0.041[*] |
| External rotation (°) | 95 ± 15 | 86 ± 18 | 0.009[*] |
| Wrist extension (°) | 2 ± 7 | 3 ± 5 | 0.728 |
| Pelvis angular velocity (°/s) | 107 ± 7 | 107 ± 9 | 0.564 |

**Notes.**
*indicates that value was found to be statistically significant.

Mean kinematics for the pre and post-test are shown in Table 5. At front-foot contact, there were no significant differences in any of the joint positions and velocities. During arm cocking, maximum internal rotation velocity was significantly higher by 232 ± 174 °/s. At ball release shoulder abduction was significantly lower by 3.0 ± 2.3 °/s and shoulder external rotation was significantly lower by 8.6 ± 5.8 °/s.

**Table 6 Kinematic data, velocity increase group.** Kinematic/biomechanical data on athletes who gained velocity in the study.

| | Pre-test (n = 9) | Post-test (n = 9) | P value for pre and post-test comparison |
|---|---|---|---|
| *Front foot contact* | | | |
| Elbow flexion (°) | 77 ± 30 | 88 ± 21 | 0.166 |
| Shoulder horizontal abduction (°) | 46 ± 18 | 51 ± 17 | 0.204 |
| Shoulder abduction (°) | 81 ± 8 | 85 ± 12 | 0.147 |
| External rotation (°) | 26 ± 27 | 21 ± 19 | 0.469 |
| Wrist extension (°) | 25 ± 21 | 23 ± 20 | 0.463 |
| Pelvis angular velocity (°/s) | 90 ± 12 | 95 ± 11 | 0.076 |
| *Arm cocking/ acceleration phase* | | | |
| Maximum pelvis angular velocity (°/s) | 717 ± 110 | 722 ± 147 | 0.862 |
| Maximum torso angular velocity (°/s) | 976 ± 116 | 1,024 ± 87 | 0.101 |
| Maximum internal rotation velocity (°/s) | 4,429 ± 453 | 4,813 ± 481 | 0.009[*] |
| Maximum elbow extension velocity (°/s)Maximum external rotation (°) | 2,166 ± 260 | 2,348 ± 327 | 0.010[*] |
| Maximum external rotation (°) | 166 ± 11 | 167 ± 11 | 0.445 |
| *Ball release* | | | |
| Elbow flexion (°) | 16 ± 6 | 16 ± 5 | 0.892 |
| Shoulder horizontal abduction (°) | −1 ± 8 | 3 ± 9 | 0.108 |
| Shoulder abduction (°) | 94 ± 7 | 91 ± 5 | 0.188 |
| External rotation (°) | 97 ± 16 | 87 ± 22 | 0.011[*] |
| Wrist extension (°) | 3 ± 8 | 5 ± 6 | 0.626 |
| Pelvis angular velocity (°/s) | 106 ± 8 | 106 ± 11 | 0.871 |

**Notes.**
[*]indicates that value was found to be statistically significant.

For the increased velocity group, there were no significant differences at front foot contact (Table 6). Maximum internal rotation velocity and maximum elbow extension velocity were significantly higher in the arm cocking phase by 385 ± 220 °/s and 182 ± 139 °/s, respectively. External rotation was significantly lower at ball release by 9.8 ± 9.2 °/s. No values were different for the velocity decrease group at front foot contact, arm cocking, or ball release (Table 7).

Maximum shoulder adduction torque was the only parameter to significantly increase (35 ± 16 °/s) during the arm cocking phase for all athletes (Table 8). For the velocity increase group, no kinetic measures were significantly different in the arm cocking phase. Maximum shoulder superior force was the only variable significantly higher (42 ± 31 °/s) in the deceleration phase (Table 9). Maximum shoulder adduction torque was the only value significantly higher (37 ± 22 °/s) in the velocity decrease group at arm cocking. Elbow anterior force (30 ± 29 °/s), elbow compressive force (95 ± 73 °/s), elbow flexion torque (11 ± 7.2 °/s), and shoulder compressive force (159 ± 122 °/s) were all significantly lower in the arm deceleration phase (Table 10).

**Table 7  Kinematic data, velocity decrease group.** Kinematic/biomechanical data on the athletes that lost velocity in the study.

| | Pre-test (n = 8) | Post-test (n = 8) | P value for pre and post-test comparison |
|---|---|---|---|
| *Front foot contact* | | | |
| Elbow flexion (°) | 106 ± 18 | 98 ± 20 | 0.067 |
| Shoulder horizontal abduction (°) | 41 ± 9 | 41 ± 17 | 0.991 |
| Shoulder abduction (°) | 85 ± 15 | 87 ± 14 | 0.561 |
| External rotation (°) | 41 ± 14 | 39 ± 24 | 0.779 |
| Wrist extension (°) | 15 ± 17 | 12 ± 18 | 0.343 |
| Pelvis angular velocity (°/s) | 94 ± 9 | 92 ± 6 | 0.461 |
| *Arm cocking/ acceleration phase* | | | |
| Maximum pelvis angular velocity (°/s) | 751 ± 100 | 720 ± 152 | 0.409 |
| Maximum torso angular velocity (°/s) | 955 ± 74 | 968 ± 117 | 0.685 |
| Maximum internal rotation velocity (°/s) | 4,638 ± 493 | 4,699 ± 633 | 0.550 |
| Maximum elbow extension velocity (°/s) | 2,302 ± 173 | 2,181 ± 327 | 0.129 |
| Maximum external rotation (°) | 170 ± 8 | 168 ± 7 | 0.155 |
| *Ball release* | | | |
| Elbow flexion (°) | 16 ± 6 | 17 ± 7 | 0.325 |
| Shoulder horizontal abduction (°) | 0 ± 6 | −1 ± 6 | 0.298 |
| Shoulder abduction (°) | 98 ± 8 | 94 ± 5 | 0.151 |
| External rotation (°) | 92 ± 14 | 85 ± 14 | 0.221 |
| Wrist extension (°) | 2 ± 7 | 1 ± 4 | 0.876 |
| Pelvis angular velocity (°/s) | 109 ± 5 | 107 ± 8 | 0.315 |

**Notes.**
*indicates that value was found to be statistically significant.

## DISCUSSION

This study investigated the effects of a baseball training program featuring weighted implements and the initial hypothesis of a significant increase in shoulder external-rotation range of motion was not supported. This was consistent for the entire subject pool as well as the sub-grouping who gained velocity, despite this phenomenon being posited as a way to enhance ball velocity (*Matsuo et al., 2001*).

It has generally been hypothesized that weighted balls work along the speed-strength spectrum. One study found significant decreases in maximal internal rotation (IR) and elbow extension (EE) velocity when throwing increasing heavier ball (*Van denTillaar & Ettema, 2011*). With a second study finding 67% of ball velocity at release could be accounted for by internal rotation and elbow extension (*Van den Tillaar & Ettema, 2004*). In our work, for the entire study sample there was a significant change in IR velocity (232 ± 174 °/s), but not EE velocity.

When our sample was broken up into those who increases and decreases velocity, we found that the velocity-increase group saw significant increases in both max IR velocity and EE velocity, whereas the velocity-decrease group saw no significant change in either metric.

**Table 8  Kinetic data, all athletes.** Kinetic/force data on all athletes in the study.

|  | Pre-test ($n = 17$) | Post-test ($n = 17$) | P values for pre and post-test comparison |
|---|---|---|---|
| *Arm cocking/ acceleration phase* | | | |
| Maximum elbow medial force ($N$) | 340 ± 60 | 350 ± 76 | 0.366 |
| Maximum elbow varus torque (N m) | 98 ± 16 | 99 ± 18 | 0.942 |
| Maximum shoulder anterior force (N) | 322 ± 196 | 299 ± 146 | 0.345 |
| Maximum shoulder horizontal adduction torque (N m) | 126 ± 39 | 123 ± 20 | 0.773 |
| Maximum shoulder internal rotation torque (N m) | 98 ± 16 | 98 ± 18 | 0.925 |
| Maximum shoulder adduction torque (N m) | 103 ± 39 | 138 ± 53 | 0.012[*] |
| *Arm deceleration phase* | | | |
| Maximum elbow anterior force (N) | 192 ± 120 | 159 ± 53 | 0.239 |
| Maximum elbow compressive force (N) | 998 ± 161 | 969 ± 167 | 0.394 |
| Maximum elbow flexion torque (N m) | 28 ± 32 | 21 ± 15 | 0.424 |
| Maximum shoulder superior force (N) | 213 ± 67 | 235 ± 80 | 0.175 |
| Maximum shoulder compressive force (N) | 1,235 ± 245 | 1,161 ± 218 | 0.072 |

Notes.
*indicates that value was found to be statistically significant.

**Table 9  Kinetic data, velocity increase group.** Kinetic/force data on the athletes that gained velocity in the study.

|  | Pre-test ($n = 9$) | Post-test ($n = 9$) | P values for pre and post-test comparison |
|---|---|---|---|
| *Arm cocking/ acceleration phase* | | | |
| Maximum elbow medial force ($N$) | 348 ± 57 | 380 ± 66 | 0.063 |
| Maximum elbow varus torque (N m) | 102 ± 15 | 106 ± 13 | 0.359 |
| Maximum shoulder anterior force (N) | 304 ± 196 | 296 ± 174 | 0.746 |
| Maximum shoulder horizontal adduction torque (N m) | 134 ± 43 | 129 ± 23 | 0.701 |
| Maximum shoulder internal rotation torque (N m) | 100 ± 15 | 105 ± 15 | 0.200 |
| Maximum shoulder adduction torque (N m) | 122 ± 35 | 155 ± 52 | 0.145 |
| *Arm deceleration phase* | | | |
| Maximum elbow anterior force (N) | 201 ± 159 | 166 ± 51 | 0.517 |
| Maximum elbow compressive force (N) | 1,019 ± 157 | 1,048 ± 172 | 0.567 |
| Maximum elbow flexion torque (N m) | 25 ± 42 | 22 ± 18 | 0.848 |
| Maximum shoulder superior force (N) | 236 ± 45 | 278 ± 81 | 0.034[*] |
| Maximum shoulder compressive force (N) | 1,248 ± 216 | 1,249 ± 213 | 0.974 |

Notes.
*indicates that value was found to be statistically significant.

**Table 10  Kinetic data, velocity decrease group.** Kinetic/force data on the athletes that lost velocity in the study.

| | Pre-test ($n = 8$) | Post-test ($n = 8$) | P values for pre and post-test comparison |
|---|---|---|---|
| *Arm cocking/ acceleration phase* | | | |
| Maximum elbow medial force (N) | 331 ± 65 | 317 ± 75 | 0.332 |
| Maximum elbow varus torque (N m) | 95 ± 17 | 90 ± 18 | 0.191 |
| Maximum shoulder anterior force (N) | 342 ± 209 | 302 ± 118 | 0.390 |
| Maximum shoulder horizontal adduction torque (N m) | 116 ± 35 | 116 ± 15 | 0.997 |
| Maximum shoulder internal rotation torque (N m) | 96 ± 17 | 90 ± 18 | 0.087 |
| Maximum shoulder adduction torque (N m) | 82 ± 34 | 119 ± 51 | 0.032[*] |
| *Arm deceleration phase* | | | |
| Maximum elbow anterior force (N) | 181 ± 61 | 151 ± 56 | 0.011[*] |
| Maximum elbow compressive force (N) | 973 ± 172 | 879 ± 114 | 0.030[*] |
| Maximum elbow flexion torque (N m) | 32 ± 17 | 21 ± 11 | 0.015[*] |
| Maximum shoulder superior force (N) | 188 ± 81 | 186 ± 42 | 0.929 |
| Maximum shoulder compressive force (N) | 1,220 ± 289 | 1,061 ± 188 | 0.044[*] |

**Notes.**
[*]indicates that value was found to be statistically significant.

There was no significant change in elbow valgus torque, derived from elbow kinematics, and the descriptive values of torque reported in this study are similar to previous studies (*Feltner & Dapena, 1986*; *Fleisig et al., 2015*).

A previous study found shoulder abduction angle at stride foot contact to be one of four variables that could explain 97% of variance in valgus stress through a regression analysis (*Werner et al., 2002*). In our study, when comparing pre- and post-training we found no significant decrease in shoulder abduction angle at stride foot contact but a significant change of abduction angle at ball release. In addition, no metrics were significantly different at front foot contact in any group.

It has been suggested that the most optimal abduction angle at release is close to 90 degrees but may vary slightly depending on the individual (*Fortenbaugh, Fleisig & Andrews, 2009*; *Matsuo et al., 2002*). The pitchers in our study saw a significant change in shoulder abduction angle at release (from 95.6 to 92.7°), moving closer to 90 degrees.

External rotation was not significantly different at front foot contact, but significantly decreased at ball release, which may be a novel finding as there is a scarcity of existing literature concerning changes in external rotation at ball release. This change was present and significant in the combined and velocity increase group.

Notably, none of our sub-groups had significant changes in elbow valgus torque or shoulder internal rotation torque as a result of the training. The increase velocity group had a significant increase in shoulder superior force, while the decrease velocity group had a significant increase in shoulder adduction torque, and significant decreases in elbow anterior force, elbow compressive force, elbow flexion torque, and shoulder compressive force.

Maximum shoulder adduction torque was significantly higher in the post-training group. Shoulder adduction torque is one of two variables related to elbow valgus torque, along with maximum internal rotation torque (*Sabick et al., 2004*). Sabick and colleagues stated that maximum shoulder adduction torque and maximum internal rotation torque were negatively correlated with elbow valgus torque, so as those two values increased, elbow valgus torque tended to decrease.

Interestingly, in our study, shoulder adduction torque only significantly increased in the group that lost velocity. The group that increased velocity had an increase in shoulder adduction torque, but it was not found to be significant.

Previous research has shown mixed results on the relationship between pelvis- and torso-angular velocity and throwing velocity, though none compared pre- and post-training periods (*Matsuo et al., 2001*; *Young, 2014*; *Dowling et al., 2016*; *Stodden et al., 2001*). Theoretically, increasing the rotational forces of the pelvis and torso allows energy to be transferred from the trunk to the throwing arm and then to the ball, which should result in higher velocities. However, our study showed no significant differences in either maximum torso angular velocity or maximum pelvis angular velocity in the pre- and post-group analysis. This remained the case even after splitting subjects into sub-groups of those who increased and decreased velocity.

These studies would also suggest that peak torso and pelvis velocities play a role in increasing velocity, but the timing is also vitally important. While the timing of peak torso and pelvis velocities was not examined in this study, further studies should examine the possible changes of constraint training and weighted balls of the timing of hip and torso rotation. Transfer of momentum during throwing is very order-dependent and typically involves a lead leg block facilitating pelvis and then trunk rotation—the peak pelvis velocity occurs before the midpoint of the time gap between stride foot contact and ball release while the peak torso velocity occurs right after said midpoint for maximum kinetic chain efficiency (*Seroyer et al., 2010*). Therefore, more research should be attempted at pre- and post-group analysis not only to look at hip and torso velocities, but also the timing difference between peak values for the two respective velocities.

Elbow flexion at ball release did not significantly change, even though a previous study found significant differences in the angle of the elbow at ball release, depending on ball weight (*Van den Tillaar & Ettema, 2004*). However, elbow flexion in our study was measured only during throws with a standardized 5-oz baseball rather than the wide gap of 0.2-kg to 0.8-kg ball weights that were employed during vadn den Tillaar and Ettema's study. As such, further research should be attempted measuring elbow flexion with different weighted baseballs.

It has also been postulated that training with weighted balls increases in external rotation, both passive and dynamic. Dynamic maximum shoulder ER has been associated with ball velocity (*Matsuo et al., 2001*; *Werner et al., 2008*), but research looking within pitcher variation found no significant association between maximum external rotation and ball velocity (*Stodden et al., 2005*). The theory holds that weighted-ball use may result in velocity gains from excess glenohumeral external rotation, which may be linked to increased elbow valgus load (*Aguinaldo & Chambers, 2009*; *Sabick et al., 2004*).

Although previous research on high-school pitchers did not find a significant correlation between passive external rotation and pitch velocity (*Keller et al., 2015*), other research did see a significant moderate correlation between passive external rotation and the degree of external rotation seen in a throw (*Miyashita et al., 2008*).

It should be noted that the biomechanical measurement of external rotation cannot be attributed only to changes of the glenohumeral joint. There can be changes in thoracic extension or scapula position that can affect measurements. In addition, the possibility of measurement error may also play a role, although the process was standardized in our work during both the pre and post testing.

Holistically, our subjects did see a passive range-of-motion increase of 1.7 degrees in the dominant arm, but the findings were not significant. Having broken up velocity into increase and decrease groups, we can see the increase group had an increase in external rotation of 2.8 degrees while the decrease velocity group saw an increase of 0.6 degrees. Interestingly, there were wide swings in the non-dominant arm external rotation. The velocity-increase group saw an increase in non-dominant external rotation of 7.8 degrees while the velocity decrease group saw a decrease of 8.6 degrees. This may bring into question what part of the changes in the dominant arm can be attributed to throwing and what parts can be attributed to non-throwing work, such as mobility or strength work, as it seems the change in non-dominant ROM came from mobility or strength work.

Although increased ER in the dominant arm was not statistically significant, it should still be considered an interesting finding since it has been suggested that humans have adapted to having more ER in order to better store elastic energy and increase power (*Roach et al., 2013*).

It has been hypothesized that training with weighted baseballs would result in negative anatomical pitching effects, such as increased ER. Our findings are interesting because the range-of-motion results reject said hypothesis of most short- and long-term range-of-motion studies.

Many of the pitchers in the study performed training days, which were either bullpens or training with weighted balls, designed to replicate high-intent pitching. The acute effects of range-of-motion on weighted balls have not been studied, but there has been research on acute changes of pitching and bullpens. It has been hypothesized that range-of-motion changes that occur in the short-term may be exacerbated over the long-term. But the research conclusions of both short- and long-term ROM changes vary.

Two studies investigating the acute effects of pitching on range of motion found a loss of shoulder internal rotation on the dominant arm that was sustained for 24 or 72 h (*Reinold et al., 2008*; *Kibler, Sciascia & Moore, 2012*).

Counter to these studies, *Freehill et al. (2014)* found that a single start resulted in no significant change in IR but rather a significant increase in passive external rotation after pitching in a game.

Another study on minor league pitching starts found both a significant decrease in internal rotation, significant gain in external rotation, and significant gain in total arm range of motion (*Case et al., 2015*). Twenty-four hours after pitching, IR returned to pre-game baseline while ER was still significantly greater.

Long-term studies examining range of motion have also found conflicting results in internal rotation and external rotation when compared to our work. *Freehill et al. (2011)* found a non-significant change in external and internal rotation. This study has a similar sample size (21 pitchers, over 29 individual seasons) compared to the 17 pitchers in our study. *Freehill et al.*'s (*2011*) study was four months in duration compared to the six weeks in our study. These pitchers also performed a capsule-stretching program during the season. Stretching programs have been seen to have positive effects on pitchers, such as reducing the likelihood of a loss in internal rotation (*Lintner et al., 2007*).

Additionally, in a follow up study, Freehill and colleagues found that preseason and postseason measurements resulted in significantly more ER, significantly less IR, and significantly less total range of motion (*Freehill et al., 2014*).

A study on baseball and softball athletes found no change in internal rotation over the course of a season but did find increased external rotation and total range of motion (*Dwelly et al., 2009*).

These long-term studies align with the acute studies, to the extent that the most common adaptations to throwing are a loss of internal rotation and a gain of external rotation, though the magnitude of change varies.

It is unknown exactly why these long-term studies differ, but it could likely be attributed to differences in the training program outside of throwing. It should be noted that none of these long-term studies found a significant increase in internal rotation in the throwing arm.

This could suggest that range of motion is a fluid measurement and hard to pin down to a discrete value for some individuals. Further research should attempt to examine if there is an acceptable range of internal and external measurements.

A loss of internal rotation may be caused by the eccentric muscle contraction that occurs in the posterior shoulder during the follow-through of pitching (*Proske & Morgan, 2001*). It is possible that no decreases were seen in our work for dominant arm internal range of motion because of the daily soft-tissue work that each pitcher completed. Although the exact causes of self-myofascial release are unknown, research has suggested SMR has positive short-term effects on range of motion without negatively affecting muscle performance (*Cheatham et al., 2015*).

As mentioned previously, the pitchers had access to instrument-assisted soft-tissue mobilization (IASTM) on an as-needed basis. Previous research on baseball players found that some acute ROM losses could be attributed to muscular/rotator-cuff stiffness, and IASTM plus stretching displayed greater gains in internal rotation than in self-stretching alone (*Bailey et al., 2015*). The gains in that study were attributed to decreased rotator-cuff stiffness and humeral retrotorsion, but not joint translation.

More specifically, one study comparing IASTM and self-stretching saw a greater increase in shoulder internal rotation and total range of motion when compared with self-stretching alone; which is similar to those found in our study (*Bailey et al., 2017*). This would suggest that soft-tissue work such as IASTM played a role in the increase in internal rotation and total range of motion that was seen in our study.

*Proske & Morgan (2001)* also hypothesized that because injuries can occur from eccentric exercise, a way to combat injury risk would be to perform an eccentric-exercise program to strengthen and, therefore, protect the muscles. Eccentric training in this program occurred while using wrist weights, j-band external and internal rotations, rebounders, and upward tosses.

Additionally, it is unlikely that the use of the Marc Pro EMS device had an effect on range of motion. It has been suggested that pitchers see reduced blood flow in their throwing arms, and the Marc Pro is used to encourage blood flow, but that would not likely result in changes in range of motion (*Laudner et al., 2014*). A study comparing different recovery techniques found that EMS resulted in a lower rating of perceived exertion and blood-lactate concentration, but no change in range-of-motion (*Warren, Szymanski & Landers, 2015*). It's unknown whether the different EMS devices used in the *Warren, Szymanski & Landers (2015)* would result in similar results.

A previous study found that performing a series of short-duration stretching/calisthenics drills (titled the Two-Out drill) resulted in short-term deficits in range of motion caused by pitching to be restored to their pre-pitching levels (*Rafael et al., 2017*). The post-throwing exercise circuit used in our study did not contain the same exercises; the exercises in our study was strength-based, not stretching/calisthenic based. However, we do show evidence that possible deficits created by throwing may return to baseline by stretching or exercise. Further studies should examine the effect that the post-throwing exercise circuit and the use of concentric and isometric exercise might have on shoulder range of motion.

A significant increase in internal rotation of the dominant arm may be seen as a positive since it has been suggested that losses of internal rotation in the throwing arm may lead to a higher risk of injury (*Wilk et al., 2011*; *Myers et al., 2006*; *Dines et al., 2009*). A study on pitchers in Japan found a relationship between more IR range of motion in their dominant arms and injury (*Sueyoshi et al., 2017*). Sueyoshi et al. included a wider range of athletes (Little League to college age) than in this study, and younger athletes have been seen to have greater IR ROM than older athletes, which may have affected the results (*Astolfi et al., 2015*). The injured group in Sueyoshi et al. also pitched in more games and more innings than the no-injury group.

The pitchers in both the pre and post measurements of our study would not qualify for either measurement of Glenohumeral Internal Rotation Deficit (GIRD,) even though the difference between non-dominant and dominant arms increased (*Burkhart, Morgan & Kibler, 2003*). This increase in the difference between internal rotation of the non-dominant and dominant arms was driven by larger increases in internal rotation range of motion in the non-dominant arm than in the dominant arm.

The concept of total range of motion (TROM) has also been introduced to examine whether differences between arms may lead to injuries (*Wilk, Meister & Andrews, 2002*). In this study, TROM saw significant increases in both the dominant and the non-dominant arm. Both arms saw larger increases in internal rotation compared to external rotation.

Furthermore, neither the pre- or post-ROM measurements qualify for either external rotation deficit (external rotation at least five degrees more in the dominant arm when compared to the non-dominant arm) or TROM deficit (when TROM of the non-dominant

arm is at least five degrees more than that the dominant arm). Pitchers with insufficient external rotation (<5 greater external rotation in throwing shoulder than non-dominant shoulder) have been seen to be more likely to have a shoulder injury (*Wilk et al., 2015*). Pitchers with deficits equal to or greater than 5 degrees in total rotation in their throwing shoulders compared to their non-dominant arms have been viewed as at higher risk of injuries (*Wilk et al., 2014*).

It's unclear from either Wilk et al. if the problem of deficits, by comparing the dominant to non-dominant arm, holds under longer term tracking and possible changes in the non-dominant arm. Even though both dominant and non-dominant TROM gained in this study, the non-dominant arm had a greater range of motion than the dominant arm post training. When examining bilateral differences in range of motion over time, researchers should take note of whether the changes are coming from the dominant or non-dominant as significant changes in range of motion in the non-dominant arm, as seen in this study, show that there can be large changes that don't come from throwing.

Humeral retroversion was not measured in this study, although this could partially explain the range-of-motion differences between the dominant and non-dominant arm (*Chant et al., 2007*). There is also research suggesting that humeral torsion adaptations occur pre-high school, suggesting that changes in this study came from soft tissue adaptations (*Oyama, Hibberd & Myers, 2013*). Further research examining range-of-motion changes and weighted-ball training should attempt to measure humeral retroversion, as well as range of motion.

This study is one of only a few that have included training programs, and as such, there is little data to compare. The throwing velocity for our group was comparable to other work, with an average initial pitching velocity of $35.1 \pm 1.8$ m/s; *Fleisig et al. (2017a)* and *Fleisig et al. (2017b)* had a group of similar amateur pitchers ($n = 25$) with an average pitching velocity of $34.2 \pm 2.0$ m/s. Fleisig et al.'s study of underweight and overweight baseball throwing showed variations in arm kinetics, variations in angular velocities, and relatively small changes in body positions. These changes could be reflective of reasonable training modalities for pitchers (*Fleisig et al., 2017a*; *Fleisig et al., 2017b*).

Our data also suggests that pitching mechanics can be changed over a six-week training period. A previous study by *Fleisig et al. (2017b)* found that pitchers can change their mechanics based off a biomechanical observation over periods of time ranging from 2-48 months. In our study, the initial screenings were not given to players with specific direction to change mechanics; the screening was purposefully observatory, yet the aforementioned significant changes in internal rotation velocity, shoulder abduction at ball release, external rotation at ball release, and shoulder abduction torque still occurred, indicating a change in individual pitching mechanics.

This paper included fourteen right-handed and three left-handed pitchers. Further research should examine the differences of weighted-ball training between right- and left-handed pitchers, as previous research has suggested differences in range of motion, humeral retroversion, and biomechanics depending on the dominant throwing arm (*Solomito, Ferreira & Nissen, 2017*; *Werner et al., 2010*; *Takenaga et al., 2017*). It is therefore

possible that pitchers should have different throwing, mobility, and strength programs depending on which arm is dominant.

## Limitations

The pitchers in this study were asked to throw as hard as comfortable on testing days. That, combined with the unfamiliarity of wearing biomechanical markers, resulted in lower velocities than what would be seen in a game or training environment.

Range-of-motion measurements were taken during the training period, so there could be unknown effects from measurements taken at different times. Range-of-motion measurements were also taken in a way that differs from other studies. Since the same subject measured every range-of-motion test, the results should be reliable but may not be directly comparable to other studies.

In addition, not every pitcher in our study had the same training background. Some had been training in-person at our facility for a few weeks while others were assessed within their first week. However, the vast majority of participants had previous experience training with weighted balls so, while hard to quantify, previous training was less of a potential confounding variable than it might have been for other research questions.

## Conclusion

This study contradicts the original hypothesis, which proposed that a 6-week training program would increase pitching velocity, arm angular velocities, joint kinetics, and arm range of motion. There were few changes comparing the pre- and post- groups, most notably there was no significant increase in elbow valgus or shoulder internal rotation torque and no significant increase in external rotation of the dominant arm. When sub-groups were created based on velocity, the velocity increase group had significant increases in internal rotation and elbow extension angular velocities.

This study contradicts the premise that weighted-implement training leads to rapid gains in shoulder external range of motion (*Reinold, 2017*). Literature on the topic of restoring shoulder internal rotation range of motion is supported (*Laudner, Sipes & Wilson, 2008*), but further research is required into individual modalities that may be contributing to these physical adaptations.

## Disclosures

It should be noted that individuals in this training program used training equipment sold out of Driveline Baseball (Kent, WA), which is owned by one of the primary authors of this study, Kyle J Boddy, and followed prescribed training programs out of the aforementioned author's published book *Hacking the Kinetic Chain*.

### Funding

The authors received no funding for this work.

## Competing Interests

All authors on this paper are employees—or partial owners—of Driveline Baseball, Inc, where this study was designed and carried out.

## Author Contributions

- Joseph A. Marsh performed the experiments, contributed reagents/materials/analysis tools, prepared figures and/or tables, authored or reviewed drafts of the paper.
- Matthew I. Wagshol analyzed the data, contributed reagents/materials/analysis tools, prepared figures and/or tables, authored or reviewed drafts of the paper.
- Kyle J. Boddy conceived and designed the experiments, performed the experiments, analyzed the data, contributed reagents/materials/analysis tools, authored or reviewed drafts of the paper, approved the final draft.
- Michael E. O'Connell performed the experiments, analyzed the data, contributed reagents/materials/analysis tools, authored or reviewed drafts of the paper, approved the final draft.
- Sam J. Briend and Kyle E. Lindley performed the experiments.
- Alex Caravan analyzed the data, contributed reagents/materials/analysis tools, prepared figures and/or tables.

## Human Ethics

The following information was supplied relating to ethical approvals (i.e., approving body and any reference numbers):

Hummingbird IRB granted ethical approval to carry out the study at the author's facilities (Hummingbird IRB #: 2017-29, Protocol WB-DLR-115).

## Data Availability

Figshare:

Main link: https://figshare.com/projects/Effects_of_a_six-week_weighted-implement_throwing_program_on_baseball_pitching_velocity_kinematics_arm_stress_and_arm_range_of_motion/34430

Boddy, Kyle (2018): All Training Data Files. figshare. Fileset. https://doi.org/10.6084/m9.figshare.6363632.v1

Boddy, Kyle (2018): Technical Data. figshare. Fileset. https://doi.org/10.6084/m9.figshare.6363629.v1

Boddy, Kyle (2018): ROM Data. figshare. Fileset. https://doi.org/10.6084/m9.figshare.6363605.v1

Boddy, Kyle (2018): C3D Files - Post Training. figshare. Fileset. https://doi.org/10.6084/m9.figshare.6363569.v1

Boddy, Kyle (2018): C3D Files - Initial Biomechanics. figshare. Fileset. https://doi.org/10.6084/m9.figshare.6363518.v1

Boddy, Kyle (2018): Tables, Figures, and Study Draft. figshare. Figure. https://doi.org/10.6084/m9.figshare.6363476.v1.

## Supplemental Information

Supplemental information for this article can be found online at http://dx.doi.org/10.7717/peerj.6003#supplemental-information.

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
