# Peer review of "Effects of a six-week weighted-implement throwing program on baseball pitching velocity, kinematics, arm stress, and arm range of motion"

_PeerJ, doi:10.7717/peerj.6003_

## Round 0.1 · original submission · Major Revisions

Dear Authors,

Your review has now been completed for the manuscript "Effects of a six-week weighted-implement throwing program on baseball pitching velocity, kinematics, arm stress, and arm range of motion". You will notice that we had three excellent and comprehensive reviews of the work by subject matter experts. I believe that these comments will help your team improve the quality of this manuscript.

The reviewers believed that the manuscript was interesting and the study relevant. However, it has been determined that a Major Revision is required. You will notice that the reviewers thought that the manuscript could be improved in a number of places. The primary concern, expressed by 2 of the reviewers (and myself), is the lack of a control group. Experimentally, this is a major issue and we want to give the authors a chance to address this. All reviewer comments can be found below. We look forward to your reply.

Regards,

·

Basic reporting

The authors support their arguments throughout with more than adequate references. There are recommendations in the attached document that suggest the authors use more sub-headings for their methods and results section. The figures are of sufficient quality. Table captions could use greater detail - a table with its caption should be able to stand alone from the document.

Experimental design

The study as it is currently written suffers a design flaw in that other studies examining the effect of weighted baseballs on velocity and injury risk have all had control groups. This study does not have a control group. The novel contribution this study provides is a set of exercises which are targeted at building a pitcher's tolerance to throwing. Theoretically, these exercises should provide a protective effect for the pitcher. The study as currently designed does not effectively deliver this message. The researcher's findings indicate there was an unexpected effect in this study - not all pitchers gained velocity. There is a post-hoc analysis of the velocity increase and decrease groups, but the main focus of this study should be centred around that (unless the researchers would agree to collecting a 6 week trial with a control group that performed weighted baseball training without these exercises - but that is neither practical or ethical).

Validity of the findings

The influence of the HTKC exercises on this type of training are significant and could be stated more prominently in the article. Kinematic data analysis appears to be done well. The fact these data were not collected in a traditional academic environment, but still had state of the art technical collection methods should be applauded and encouraged.

Additional comments

Very important work with the rise of weighted baseball training. It's also very important to shift some of the research being done from purely academic labs to the places where athletes are actually going to get better. Sometimes there are things overstated that don't make that big of an impact to your study, and sometimes there are things understated that would make a huge impact. I would be happy to review this research again, as I see it as being very important to the industry and sports science.

Reviewer 2 ·

Basic reporting

The authors presented a well written manuscript on a very topical subject that requires much more experimental study (especially in the open literature). For their efforts, the authors must be commended.

With respect to basic reporting, there are only minor reviews suggested at this time. They are as follows:

Abstract: In the results section of the abstract, can you clarify which experimental group the outcome variables were higher for to make it clear for a reader that only reads the abstract?

Lines 271-275: This appears to be a planned post hoc analysis and therefore should be described in the methods section before it is introduced in the discussion

Experimental design

From an experimental design perspective, the study presented in the mansucript has some severe limitations that must be addressed. The authors are attempting to study the effects of a "weighted-impleneted throwing program" on velocity and biomechanical outcome variables. The gold standard for this study would be a RCT. The presented study is not that type of study. The reviewer understands there are often logistical limiatations (or other constraints) that cannot be easily overcome to produce a gold standard study, and the reviewer concedes that it is still often valuable to present other studies in the literatrue. With that being said, there are options the authors could explore to improve the design of their study.

Specifically:

Lines 91-98: In terms of exclusion criteria, where participants asked about previous training modalities they have utilized? Was this training modality novel to all participants or had some undergone similar types of training previously? This comment is designed to tease out whether or not previous training could be a confounding factor.

Knowing more about each individual participants training history would help provide context for the pre- post- changes being statistically tested in this study. Although they would not be able to technically serve as their own within subject controls, the study results could be presented and interpreted with much greater confidence.

From a methodological perspective, one last concern remains:

Lines 157-161: For the reader not familiar with the new Fleisig method cited as Fleisig et al. 2017, does this biomechanical marker comply with ISB suggested model (Wu et al. 2005)? Based on the extensive marker set described in Lines 131-140, the biomechanical model may be assumed, but an explicit statement on the model is still beneficial for the reader.

Validity of the findings

The validity of the finidngs are difficult to comment on given the limitations in experimental design.

Lines 526-534: This comment is a fair comment and in the strength and conditioning literature this problem is somewhat systematic. One suggestion this reviewer can make is for the authors to go back to their participant pool (if avaiable) and get detailed training histories for the weeks or months leading up to the study. This will help provide context for the results and improve the validity of the findings.

Lines 567-568: Small sample size is a difficult problem to overcome in any human experimentation that involves as extensive an intervention as that described in this manuscript. One suggestion to increase statistical power is to include control matches in the study.

Additional comments

Two summeraize, this reviewer's two major suggestions for the authors are:

1. Include a control matched group of participants that use a non weighted-implemented throwing program so that the true effects of the weighted implemented throwing program can be properly teased out. (Understood this may not be possible)

2. Alternatively, go back to the participants of the study and survey the participants with respect to their training prior to the study. This will provide better context for the results and discussion and improve the validity of the findings.

·

Basic reporting

see attached document

Experimental design

see attached document

Validity of the findings

no comments.

Additional comments

Well done study and impressive literature review for this paper.

---

## Round 0.2 · Minor Revisions

Dear Authors,

Thank you for such a detailed and careful response to the previous revision request. I have now received further reviews from two experts in areas with particular relevance to your paper. You will see from the reviewer comments that this work has greatly improved. All reviews are generally positive and supportive of this work.

I have also read your manuscript and the reviewers’ comments. I am also in support of this work, but I have concluded that your manuscript is not yet acceptable for publication in PeerJ. However, I encourage you to revise and resubmit your manuscript.

I have read the manuscript carefully and although the reviewers are mostly satisfied with the current form, there are still considerable places that need further attention. Overall, the writing needs to be more scientific in nature. Some methods require further details and the discussion needs some re-organization.

I hope that you are able to address these concerns as I believe that the result will be a stronger contribution to the journal. You will find the reviewer comments below and I have attached a marked up PDF with my own comments/edits.

Regards,

·

Basic reporting

Limitations/Conclusions
Reinold reference – either include first initial in reference, or don’t include the initial – consistency throughout.

Table 1: Some inconsistencies in this table. This table has some inconsistencies. We know the n = 19 - the header on first column should be Age (Years), and following columns switch between all caps or lower case for units (CM vs. cm). Be consistent with these.

Experimental design

Line 72: Suggestion for this paragraph leading into the purpose: I think this more clearly states what the intentions of this study are and makes no inferences to an RCT: Feel free to disagree or modify, but I feel this clarifies your purpose, without indicating that we’re doing some sort of an RCT. Purely looking at the changes that occur during this training alongside the manual therapy and HTKC program, and we can do control group studies in the future.

Driveline baseball (Seattle, Washington, USA) has developed weighted baseball training programs, which have been used by many professional and collegiate pitchers. Those pitchers who completed the weighted implement training programs Driveline Baseball’s summer training programs have on average increased pitching velocity 2.7 MPH in 2016 and 3.3 MPH in 2017 (Driveline Baseball, 2016 and 2017). However, there remains no conclusive evidence explaining the mechanism of the velocity gains, and research indicates the phenomenon of weighted-ball training increasing “arm strength” may be incorrect (Cressey, 2013).

Line 84: Purpose Statement: Suggest rephrasing:
The purpose of this study is to examine changes in kinetics, kinematics, and performance, occurring in pitchers who complete weighted implement and individualized training routines geared around combating the negative effects of throwing. We hypothesize the previously described program will increase external rotation, ball velocity, and elbow varus torque.

Line 90: Use of sub-headings is helpful to break up a detailed methods section like this one. I think you can break the methods down even further, particularly after line 112.
• Range of motion testing
• Kinematics
• Pitcher testing preparation
• Data Analysis

• Training Methods
o Warm up
o Resistance training
o Plyometric training
o Long toss
o Etc

Line 316: Define what “cleaning data means”. Grubbs Analysis and data replacement?

Validity of the findings

Results section: wherever there are values reported, present variance as well as means. +/- SD or SE.

Additional comments

Authors made significant improvements in the manuscript. Suggestions included are to just strengthen the manuscript - it is acceptable for publication.

Reviewer 2 ·

Basic reporting

The authors have addressed each of this reviewer's concerns with respect to basic reporting.

Experimental design

The authors were not able to address the authors primary concern with respect to including a true control matched group. Nor were the authors able to re-connect with participants to include detailed training histories prior to the study. With that being said, the authors were able to provide a general description of the participants' training prior to the study and have been able to acknowledge the limitation in their experimental design.

Validity of the findings

Given the addition of a general description of the participants' training history and the acknowledged limitations of the experimental design, the study is now better framed to provide valid findings within the limited scope.

Additional comments

None.

---

## Round 0.3 · accepted · Accept

Thank you for a much improved manuscript and for being patient with my editorial review. I have been through your manuscript again and feel that it can now be published.

I do have a few more comments that I hope you will consider in good faith prior to publication. In particular, the discussion is very long and somewhat disjointed. I would suggest shortening when possible (see my comments in the attached PDF) and attempt to better section the discussion into major topics, rather than scattered paragraphs at times.

Thanks again for your contribution to PeerJ and I look forward to seeing the final published version. Great work!